# Multi-Parameter Analysis of Disseminated Tumor Cells (DTCs) in Early Breast Cancer Patients with Hormone-Receptor-Positive Tumors

**DOI:** 10.3390/cancers15030568

**Published:** 2023-01-17

**Authors:** Theresa König, Senol Dogan, Anne Kathrin Höhn, Laura Weydandt, Bahriye Aktas, Ivonne Nel

**Affiliations:** 1Department of Gynecology, Medical Center, University of Leipzig, 04103 Leipzig, Germany; 2Department of Pathology, Medical Center, University of Leipzig, 04103 Leipzig, Germany

**Keywords:** breast cancer, disseminated tumor cells, bone marrow, hormone receptor, HER2, dormancy, proliferation

## Abstract

**Simple Summary:**

Despite successful treatment of the primary tumor, recurrence occurs in about 30% of breast cancer patients. This could be due to disseminated tumor cells (DTCs), which split off during the early stages, metastasize to the bone marrow and cause relapse years after initial diagnosis, preferentially in hormone-receptor-positive patients. Currently, standardized DTC analysis is based on the detection of epithelial cells in the bone marrow. In this study, we established a sequential multi-parameter staining procedure to investigate phenotypical and therapy-related features of these rare cells. We found distinct receptor profiles that enormously differed from the primary tumor tissue and might have clinical implications. Especially, patients with luminal A tumors revealed DTCs with potential therapeutic targets. We analyzed particular non-epithelial DTC subgroups and clusters that were never described before. Our findings indicate that characterization rather than quantification of DTCs might be relevant for prognosis and treatment decisions.

**Abstract:**

Background: Patients with hormone-receptor-positive (HR+) breast cancer are at increased risk for late recurrence. One reason might be disseminated tumor cells (DTCs), which split off in the early stages of the disease and metastasize into the bone marrow (BM). Methods: We developed a novel multi-parameter immunofluorescence staining protocol using releasable and bleachable antibody–fluorochrome-conjugates. This sequential procedure enabled us to analyze six distinct phenotypical and therapy-related markers on the same DTC. We characterized BM aspirates from 29 patients with a HR+ tumor and a known positive DTC status—based on the standardized detection of epithelial cells in BM. Results: Using the immunofluorescence staining, a total of 153 DTCs were detected. Luminal A patients revealed a higher DTC count compared with luminal B. The majority of the detected DTCs were CK-positive (128/153). However, in 16 of 17 luminal A patients we found HER2-positive DTCs. We detected CK-negative DTCs (25/153) in 12 of 29 patients. Of those cells, 76% were Ki67-positive and 68% were HER2-positive. Moreover, we detected DTC clusters consisting of mixed characteristics in 6 of 29 patients. Conclusions: Using sequential multi-parameter imaging made it possible to identify distinct DTC profiles not solely based on epithelial features. Our findings indicate that characterization rather than quantification of DTCs might be relevant for treatment decisions.

## 1. Introduction

Even though primary breast cancer can be treated successfully, metastatic relapse might occur months to years after initial diagnosis, causing recurrence-related death in up to 30% of patients [1]. One reason might be hematogenous spread during the early disease stages, when tumor cells dissolve from the primary tumor and travel along the blood stream and lymphatic system into distant organs. In breast cancer, disseminated tumor cells (DTCs) preferentially migrate into the bone marrow, where they become dormant [2]. In this steady state, they remain viable but not proliferative. Thus, they are not affected by chemotherapy [3]. That DTCs may serve as independent prognostic markers associated with impaired survival has been shown in various studies [4,5,6,7]. The mechanisms triggering DTCs to re-awaken and cause recurrence years after primary diagnosis are not fully understood yet. Numerous methods were performed to analyze DTCs in the bone marrow of patients with breast cancer, and different antibodies were employed [8,9,10]. Since DTCs are very rare among hematopoietic cells in the bone marrow, their detection and characterization remain challenging. For the time being, there is no clinical test to stratify patients at elevated risk for recurrence based on their DTC profile at primary diagnosis. Although bisphosphonate treatment has been shown to effectively eliminate DTCs in the bone marrow of breast cancer patients [11], there is no therapy directly targeting DTCs. Throughout the disease and therapy courses, tumor cells are able to change their phenotype as well as functional and mechanical properties, leading to tremendous heterogeneity and thus discordance between the primary tumor and DTCs [12,13,14,15]. Currently, treatment decisions are based on the properties of the primary tumor gained as a tissue biopsy at diagnosis. Here, molecular properties are crucial, since tumors can be divided into distinct subtypes, namely luminal A, luminal B, HER2-enriched and triple-negative breast cancer [16]. Patients with luminal A tumors are hormone-receptor-positive (HR+), meaning they show estrogen receptor (ER) and progesterone receptor (PR) expression and a low proliferation index in terms of low Ki67 expression. Usually, they benefit from endocrine therapy. Luminal B tumors are defined as HR+ and either HER2+ or HER2− plus high Ki67 index. In addition to endocrine therapy, patients can be treated with targeted treatment against HER2 and chemotherapy. Targeted therapy with the monoclonal antibody Trastuzumab directed against HER2 is standard treatment for patients with HER2-enriched tumors, whereas patients with triple-negative tumors often receive chemotherapy due to them lacking targets but having a high proliferation index. Various studies discovered DTCs that were expressing HER2 in the bone marrow of patients with HER2-negative primary tumors, however [17,18,19]. Distinct DTC subpopulations might even be able to evade targeted or systemic therapy and promote recurrence. Further, the reversible process of epithelial-to-mesenchymal transition (EMT) is playing a role during metastasis and was reported to be associated with increased tumor aggressiveness, invasiveness, and chemo-resistance [20,21]. It has already been reported that tumor cells that disseminate from the primary tumor are to likely go through EMT, which leads to phenotypical and molecular changes and renders epithelial cells more motile, flexible, invasive, and capable of metastasis [22]. Hence, tumor cells that went through EMT might express a fundamentally different subset of epithelial and nuclear markers compared with the primary tumor. Thus, DTCs with mesenchymal or EMT-like features might be missed when using detection methods solely based on epithelial markers such as pan-cytokeratin (Pan-CK). In this study, we developed a sequential multi-parameter imaging method based on immunofluorescence staining that allowed the investigation of Pan-CK as an epithelial marker, CD133 for stemness, Ki67 for cell proliferation, and ER as well as HER2 as therapy-related markers on the same DTC. We scrutinized DTC subpopulations among a HR+ cohort and intended to find clinical associations. 

## 2. Materials and Methods

### 2.1. Study Population and Informed Consent

The study was conducted at the Department of Gynecology at the University Hospital Leipzig, Germany. After agreeing and signing a written informed consent in accordance with the requirements of our institution’s board of ethics (internal reference number: No. 216/18-ek), bone marrow aspirates from patients with histopathologically confirmed breast cancer were sampled during surgery. Clinico-pathological patient data were collected from medical records, and patient characteristics are listed in Table 1. In total, 29 patients were enrolled in the study. At diagnosis, core needle biopsies of the breast tumor were obtained. The median Ki67 proliferation rate was 18.5% throughout the cohort (ranging from 1% to 80%). All patients were positive for estrogen receptor (ER) and progesterone receptor (PR). Additionally, 6 (21%) patients were positive (IHC DAKO Score 3+) for the human epidermal growth factor receptor 2 (HER2). Among the cohort, the median age at sample withdrawal was 54.8 years, ranging from 33 to 85 years. Almost all patients (n = 25; 86%) presented with early-stage breast cancer and about a third (n = 11; 38%) received neoadjuvant therapy. Of those, 8 patients received neoadjuvant endocrine therapy and 3 patients received neoadjuvant chemotherapy combined with targeted therapy against HER2. Local lymph node involvement was found postoperatively in 12 of 29 patients (41%), and one of the patients had evidence of distant metastatic disease. The majority of the patients (n = 18; 62%) presented with T1 tumors. Table 1 shows the characteristics of the patients included in this cohort.

### 2.2. Bone Marrow Aspirates

Bone marrow aspirates were collected from the anterior iliac crest during surgery. After density gradient centrifugation, cell suspensions were transferred onto glass slides (1 × 10^6^ cells per slide) using a cytospin centrifuge, fixed with ice-cold methanol and stored at 4 °C until subjected to immunocytochemical staining. We prepared 8 slides per bone marrow aspirate. Remaining bone marrow cell suspensions were stored in liquid nitrogen for further investigation later on. In this study, bone marrow aspirates of 29 patients were included who met the criteria of a primary ER-positive breast cancer as well as prior positive detection of DTCs in the same aspirated bone marrow (4 of 8 slides) using a standardized brightfield detection method with an antibody directed against Pan-CK and labelled with alkaline phosphatase (Cat. No.130-090-462, Miltenyi Biotech), based on the protocol established in our laboratory [23]. Hence, DTCs of each included bone marrow aspirate were quantified with both a standard brightfield method and the novel sequential multiparameter staining. As a positive control with each run, we used reference slides with a mix of bone marrow cells and ZR75-1 (breast cancer cells) plus T98G (glioblastoma cells) which were prepared, stored, and fixed in the same manner. 

### 2.3. Sequential Immunofluorescence Staining and DTC Imaging

Knowingly that there are a great variety of commonly used phenotype-related markers available, we developed a sequential multi-parameter imaging method based on immunofluorescence staining that allowed the investigation of Pan-CK as an epithelial marker, CD133 for stemness, Ki67 for cell proliferation and ER as well as HER2 as therapy-related markers on the same DTC.

Per patient, we analyzed 2 × 10^6^ prepared bone marrow cells using a sequential multi-parameter staining procedure applying antibodies with releasable or bleachable fluorochrome conjugates, respectively. The use of the Aperio Versa microscope-based scanning system (Leica Biosystems), which allows annotation of single cells and the position thereof, plus the use of releasing enzymes as well as photobleaching steps, made it possible to stain the same DTC multiple times against 6 different markers in total.

Respectively, two markers (Pan-CK & CD133 and HER2 & Ki67) were stained simultaneously before the samples were covered with mounting medium containing DAPI for DNA staining in the nucleus. Then, images of the DTCs were acquired. Thereafter, the two respective markers were released (Pan-CK & CD133) or bleached (HER2 & Ki67) to enable further immunofluorescence staining. In brief, the staining was performed as follows: washing of the slides first in TBS-T for 10 min at room temperature, then for 30 min in 5% BSA in TBS-T to block for cross-reactions at room temperature. This was followed by incubation with conjugated antibodies for 60 min at room temperature. In the first round of staining, anti-human Pan-Cytokeratin APC-conjugated REAdye_lease^TM^ antibody (Clone REAL648, Cat. No. 130-123-091, Miltenyi Biotech) and anti-human CD133/1 PE-conjugated Antibody REAlease^®^ (Clone REAL233, Cat. No. 130-118-061, Miltenyi Biotech) were used. 

To detect the secondary fluorescence emitted from the fluorochromes, microscopic scans of the stained slides were acquired using band pass excitation filters D for DAPI (emission 457 nm), Y3 for phycoerythrin (PE; emission 574 nm) and Y5 for allophycocyanin (APC; emission 660 nm) at 20× magnification before uncovering the slides and releasing the antibody–fluorochrome complex from the cells using the REAlease Release Reagent (Cat. No. 130-120-675, Miltenyi Biotech). In the second round of staining, anti-human/mouse Ki67 antibody conjugated with PE (REAfinity^TM^, Clone REA183, Cat No. 130-120-417, Miltenyi Biotech), and, for HER2 detection, anti-human ErbB-2 (CD340) antibody conjugated with APC (REAfinity^TM^, Clone REA1232, Cat. No. 130-124-474, Miltenyi Biotech) were applied. After image acquisition, the stained slides were uncovered and photobleached by scanning with an 1450 ms exposure time for the Y3 filter canal, as well as a 4000 ms exposure for the Y5 filter canal. This ensured full bleaching and hence destruction of the previously stained fluorochromes PE and APC, which have excitation peaks of 566 nm and 651 nm wavelength, respectively. Bleaching was done over a period of approximately 25 min per slide while preventing drying of the slides, thus preparing the cells for a final staining round. For the third staining, the estrogen receptor alpha antibody, Cy5-conjugated (Cat. No. bs-0725R-Cy5, Bioss Antibodies) and anti-human/mouse CD325 (N-Cadherin) –PE antibody (Clone 8C11, Cat. No. 130-116-170, Miltenyi Biotech) were employed. For every patient sample, we acquired three whole slide images per channel (DAPI, Y3 and Y5), resulting in nine images per patient sample. For DTC analysis and quantification, the three resulting images of every patient sample in one of the three channels were aligned via the “ImageScope” software of the Leica Scanning microscope. Hence, using overlays of the three images of individual DTC candidates in each channel allowed the detection of distinct cellular profiles by positivity or negativity for Pan-CK, CD133, HER2, Ki67 N-Cadherin and ER.

### 2.4. Cell Culture for Reference Slides

Breast cancer cells from the hormone-receptor-positive cell lines ZR75-1 and the fibroblast-like glioblastoma cell line T98G were used as positive controls. The cells were maintained under standard conditions at 37 °C in a 95% air and 5% CO_2_ atmosphere in Dulbecco’s Modified Eagle Medium (DMEM) containing 4.5 g/L glucose and l-glutamine (Cat. No. FG 0435, Biochrom) supplemented with 10% fetal bovine serum (Cat. No. S 0615, Biochrom) and 100 U/mL penicillin/streptomycin. They were harvested using trypsin/EDTA solution, re-suspended in media, and left in the incubator for 15 min for cell surface recovery. After washing in phosphate-buffered saline (PBS), they were counted using the Countstar (INTAS Science Imaging Instruments; Germany). Then 10,000 ZR75-1 cells and 10,000 T98G cells were spiked into bone marrow, resulting in a concentration of 20,000 cell culture cells per 1 million bone marrow cells. Cells were spun onto slides using a cytospin centrifuge before fixation with ice cold methanol for 5 min.

### 2.5. Statistical Analysis

Spearman Rho rank correlation, Pearson correlation, and descriptive statistics were used to examine associations between DTC subtypes and the clinical characteristics of the patients. DTC profiles were divided into distinct groups, such as CK+ and CK-, to find correlations among various markers. In addition, patient data were divided according to characteristics such as age, menopausal status, primary tumor subtype and neoadjuvant therapy to analyze differences in DTC profiles. The R statistics program and R-based Jamovi program were used, and statistical significance was set at *p* ≤ 0.05 [24,25].

## 3. Results

### 3.1. Staining of Reference Slides Using Sequential Immunofluorescence

For the investigation of cellular subtypes, a multi-staining method was required to detect epithelial, mesenchymal, stem-cell-like, and therapeutically relevant markers. Therefore, we established sequential immunofluorescence staining as described above for DTC subtype detection in HR+ breast cancer patients (Figure 1). A mixture of bone marrow cells spiked with breast cancer cells from the cell lines ZR75-1 (epithelial phenotype) and the glioblastoma cell line T98G (mesenchymal phenotype) was used as a positive control and negative control for each run. The epithelial ZR75-1 breast cancer cell stained positive for Pan-CK, HER2, Ki67, and ER, whereas the T98G glioblastoma cell line was positive for CD133, Ki67, and N-cadherin and remained negative for Pan-CK, HER2, and ER. Hematopoietic cells were negative for all the above-mentioned markers, displaying a positive nuclear staining using DAPI, however (Table 2).

After the first round of staining with antibodies against Pan-CK and CD133, the cells were treated with REAlease Release Reagent to release the REAlease Complex (recombinantly engineered antibody fragments with low epitope binding affinities and no binding to Fc receptors, Cat. No. 130-120-675, Miltenyi Biotech). To ensure the effectiveness of the procedure, we analyzed the average concentration of dye in cellular compartments in the positive controls pre- and post-release. The Aperio Cellular IF Algorithm was employed to calculate the dye intensity based on integrated intensity of pixels and percentage coverage of the color within and across cellular compartments. Thus, we could confirm a 90.7% and 94.2% reduction of dye intensity in the Y3 and Y5 channels, respectively (Figure 2A). 

After the second round of staining with antibodies against HER2 and Ki67, the cells underwent photobleaching using a scanning run with high light exposure times in the Y3 and Y5 channels to quench secondary fluorescence. Again, to ensure the effectiveness of bleaching, we analyzed dye intensity with the Aperio Cellular IF Algorithm and could confirm a signal reduction by 80.7% and 84.6% in the Y3 and Y5 channels, respectively (Figure 2B). 

In the third round of staining, cells were labelled with antibodies against ER and N-cadherin. After image acquisition, detection, and quantification, DTC subtyping was performed manually using the Aperio Versa Image Scope program (Figure 3 and Figure 4). 

### 3.2. Identification and Quantification of DTC-Subtypes in HR+ Patients

When examining bone marrow samples from breast cancer patients, cells that showed a positive nuclear staining with DAPI and a positive staining against Pan-CK, ER, HER2, or multiple markers at once were captured and considered as tumor cells. Cells only showing positive staining for markers CD133, Ki67, or N-cadherin (Ncad), respectively, were not considered as tumor cells, due to the possibility of hematopoietic cells displaying those markers. In contrast to immunohistochemical Ki67 staining in tissue, we did not apply any scoring. Based on the fluorescent signal, a cell was either positive or negative for Ki67 in this study. Figure 5 shows a representative image of a DTC from a patient sample exhibiting the profile CK+CD133+HER2+Ki67+Ncad+. 

In total, we detected 153 DTCs using the sequential staining approach, resulting in a median number of 5 DTCs per patient. The majority of patients showed cell counts of one to two DTCs per 2 × 10^6^ bone marrow cells; nine patients revealed DTC counts of six and higher (Figure 6). 

### 3.3. DTC Subpopulations among Luminal A and Luminal B Tumors

Among the cohort, 17 patients presented with a luminal A tumor, while 12 patients presented with a luminal B tumor. Patients with luminal A tumors revealed a higher DTC count compared with luminal B (6 of 12 HER2-positive; median 5 vs. 2 DTCs per patient).

In total, we detected 128 DTCs that were positive for CK, either alone or in combination with other markers. The majority of the CK+ cells occurred among patients with luminal A tumors (Table 3). Further, we detected 56 DTCs that were HER2-positive. Noteworthily, 45 of them were found in patients with luminal A tumors (mean 2.65 cells). Remarkably, of the 17 patients with luminal A tumors, 16 had HER2+ DTCs. We detected 87 DTCs that were Ki67-positive and predominantly found in patients with luminal A tumors (62/87 cells). However, only 10 of 17 patients with luminal A tumors had ER-positive DTCs. In total, 24 DTCs were ER-positive in any combination with the other makers or alone (Table 3).

We detected 41 CK+ cells that were only positive for CK (mean 1.86), whereof 33 occurred among patients with luminal A tumors (mean 1.94), 1 DTC among patients with luminal B/HER2− tumors (mean 0.17), and 7 cells (mean 1.17) among patients with luminal B/HER2+ primary tumors. We found 25 cells that were CK+CD133+Ki67+, whereof 19 (mean 1.12) were found in patients with luminal A tumors and 6 (mean 1.0) in patients with luminal B/HER2− tumors. Interestingly, we discovered only four DTCs that were CD133+HER2+Ki67+, but all of the four patients were diagnosed with luminal A tumors. We detected 12 CK+CD133+HER2+Ki67+Ncad+ DTCs among nine patients, whereof seven had luminal A (mean 0.47), one had luminal B/HER2− and the other one had luminal B/HER2+ tumors (mean 0.33 each). Further, we revealed six CK+CD133+HER2+Ki67+ER+Ncad+ cells among four patients (3× luminal A and 1× luminal B/HER2+). Moreover, we found six CK+CD133+ DTCs among four patients (3× luminal A, 1× luminal B/HER2−) and 7 CK+HER2+ DTCs among 6 patients with luminal A tumors. Remarkably, we did not detect any CK+CD133+/HER+ cells, but we found nine CK+CD133+HER2+Ki67+ cells (among nine patients) and only three CK+Ki67+ cells (three patients all subtypes), indicating that Ki67 appears to occur together with HER2. Interestingly, eight of the nine patients had HER2-negative primary tumors (6× luminal A, 2× luminal B/HER2−). We found five CK-negative but HER2+ DTCs among three patients with luminal A primary tumors, and three more patients with luminal A tumors revealed CK-negative but CD133+HER2+Ki67+ cells in the bone marrow (n = 5 cells). 

Looking at the descriptives, among patients with luminal A and luminal B/HER2− primary tumors, we discovered an elevated number of Ki67-positive DTCs (mean 3.65 rsp. 2.83) and ER-positive DTCs (mean 0.88 rsp. 0.83) compared with luminal B/HER2+ (mean 1.33 for Ki67+ and 0.66 for CK+). The amount of CK-positive DTCs was increased in samples derived from patients with luminal A primary tumors (mean 5.59) compared with luminal B/HER2− (mean 2.83) and luminal B/HER2+ (mean 2.67). 

In six bone marrow samples of patients with luminal A tumors we detected CK+HER2+ cells, indicating a receptor discordance between primary tumor tissue and DTCs. Further, only three patients displayed CK+ER+ DTCs, although the entire cohort was diagnosed with hormone-receptor-positive tumors. 

We detected only one CK+Ncad+ cell in only one patient and one CK+CD133+Ki67+ER+ in another patient (both luminal B/HER2+), but we found eight CK+CD133+Ki67+Ncad+ cells in six patients (luminal A and luminal B/HER2−). Five of them had CK+CD133+Ki67+ cells, suggesting that the occurrence of N-cadherin might be associated with CD133 and Ki67. Noteworthily, in two patients we found CK-negative DTCs with the profile CD133+Ki67+ER+Ncad+ cells. Here, Pearson correlation revealed that among CK-negative DTC subpopulations HER2+ cells were associated with CD133+Ki67+Ncad+ DTCs (r = 0.9). Among CK-positive DTCs, CK+HER2+Ncad+ cells were correlated with CK+ER+ (r = 1.00) and CK+ cells were associated with CK+CD133+Ki67+/Ncad+ (r = 0.78). Spearman–Rho correlation showed that the sum of DTCs detected was significantly higher in patients with the luminal A subtype compared with luminal B/HER2− or luminal B/HER2+ subtypes (r = −0.42, *p* = 0.025). The luminal A subtype also had a higher likelihood of HER2+ DTCs being detected than luminal B/HER2− or luminal B/HER2+ (r = −0.59, *p* = 0.001), suggesting discordance between DTCs and the primary tumor profile. 

### 3.4. Detection of CK-Positive and CK-Negative DTC Subpopulations

The majority of detected DTCs (128 of 153) were CK-positive (Figure 7A). The most common profile in CK-positive cells was CK+ alone (mean 1.86), followed by CK+CD133+Ki67+ (mean 1.07). Remarkably, we did not detect any DTCs with the profile CK+CD133+HER2+. In seven out of eight profiles, CD133 was only present if Ki67 was also positive, suggesting a connection between these two markers.

In total, 25 of the 153 detected DTCs were CK-negative (Figure 7B). We found CK-negative cells in 12 of the 29 patients (41%). The most frequently occurring profiles in these patients were HER2+ alone and CD133+HER2+Ki67+Ncad+ (mean 0.31 rsp. 0.31), closely followed by CD133+HER2+Ki67+ and CD133+Ki67+ER+Ncad+ (mean 0.25 rsp. 0.26). Of these CK-negative DTCs, 76% showed Ki67 positivity (Table 4). In contrast, Spearman correlation revealed that only 53% of CK-positive DTCs showed Ki67 positivity (r = −0.373, *p* = 0.055). Of the CK-negative DTCs, 68% were HER2-positive, compared with only 30% HER2 positivity in the CK-positive DTCs. Almost half (48%) of the CK-negative DTCs showed positivity for N-cadherin, while only 22% of the CK-positive DTCs were positive for N-cadherin.

### 3.5. Classification of DTCs According to Phenotypes

To define a predominant DTC phenotype for each patient, the markers CK, CD133 and N-cadherin were used to assign each patient’s DTCs an epithelial, mesenchymal, stem-cell-like, or mixed phenotype. Cells expressing CK positivity were counted as epithelial. Cells showing CD133 positivity were counted as stem-cell-like, while cells expressing N-cadherin positivity were counted as mesenchymal. Cells expressing positivity for multiple of these markers counted in multiple categories. The cell counts in these categories were used to calculate a ratio (absolute DTC count/phenotype DTC count) to determine the predominant phenotype for each patient. If the ratio was the same for two or more phenotypes, the patient was assigned a “mixed” phenotype. In this cohort, the most common DTC phenotype was an epithelial phenotype, with 55% of patients expressing more epithelial than any other cells (Figure 8). However, almost a third of the patients expressed a mixed phenotype (28%). While 14% showed a predominantly stem-cell-like phenotype in their DTCs, only 3% of patients showed DTCs that were predominantly mesenchymal. 

### 3.6. DTC-Subtypes and Clinical Data

Spearman’s Rho correlation was used to find connections between clinical parameters and DTC subtypes. We revealed a significant correlation between the age of patients and the likelihood of HER2+ DTCs in the bone marrow (r = 0.45, *p* = 0.014). Although not significant, there was a correlation between the sum of HER2+ DTCs and higher age (r = 0.32). The presence of HER2+ DTCs in the bone marrow was correlated to higher numbers of Ki67+ DTCs (r = 0.37, *p* = 0.049), suggesting a connection between the expression of these two markers. As expected from a methodological point of view, a higher number of CK+ DTCs was correlated to a higher number of HER2+ DTCs (r = 0.52, *p* = 0.004). ER positivity of DTCs correlated negatively with the likelihood of Ki67+ DTCs being detected in the bone marrow (r = −0.54, *p* = 0.003). ER+ DTCs also occurred more often in patients with a stem-cell-like or mixed DTC phenotype (r = 0.59, *p* = 0.001). In contrast, Ki67+ DTCs were more likely in patients with an epithelial or mesenchymal phenotype (r = −0.38, *p* = 0.05). Patients that did not receive therapy were more likely to display DTCs with a stem-cell-like or mixed phenotype compared with patients who had received neoadjuvant therapy (r = −0.387, *p* = 0.038). Patients receiving neoadjuvant chemotherapy or targeted therapy were more likely to express an epithelial phenotype (r = −0.413, *p* = 0.026). 

### 3.7. Receptor Discordance between DTCs and Tumor Tissue Biopsy

Between DTCs and primary tumor biopsy tissue obtained at diagnosis, we observed a receptor discordance for HER2 in 79% (n = 23) and for ER in 41% (n = 12) of the patients. Receptor discordance was defined either as the detection of at least one DTC that expressed HER2 when the primary tumor did not or as no detection of ER-respective HER2-positive DTCs when the primary tumor was immunohistochemically positive for either of those markers. 

Almost all (16 of 17) patients presenting a luminal A subtype in the tumor biopsy showed at least one DTC-expressing HER2 positivity and 41% showed a discordance for ER in this subtype, meaning that ER+ DTCs were expected but not present. In the luminal B/HER2− subtype, only 50% (3 of 6) showed HER2 discordance, meaning that they presented HER2+ DTCs, and 33% (2 of 6) showed an ER discordance by lacking ER+ DTCs. Furthermore, patients with luminal B/ HER2+ tumors showed HER2-negative DTCs in four out of six cases (67%) and ER discordance in 50% (three of six patients). Prior to surgical therapy and bone marrow aspiration, 11 out of 29 patients (38%) received neoadjuvant therapy. Of those, 82% (n = 9) showed a discordance for HER2 and 36% (n = 4) showed a discordance for ER (Table 5). In comparison, 18 patients did not receive therapy and showed similar discordance rates of 78% (n = 14) for HER2 and 44% (n = 8) for ER.

In this cohort, 8 out of 29 patients received neoadjuvant endocrine therapy (Table 5). Of these, 25% showed a discordance for ER between the tumor biopsy and DTCs, meaning ER+ DTCs were not present despite the ER positivity in the primary tumor. In contrast, 21 patients did not receive neoadjuvant endocrine therapy. Of those patients, 48% showed a discordance for ER. 

Neoadjuvant targeted therapy with trastuzumab or pertuzumab in combination with chemotherapy was received by three patients, all of which had luminal B/HER2+ primary tumors. Two-thirds of them showed a HER2 discordance, because they presented with HER2-negative DTCs. These patients did not present ER-positive DTCs either. Among the cohort, 26 patients did not receive targeted therapy. However, 81% of them were HER2-discordant to the HER2-negative primary tumor, presenting HER2-positive DTCs, and thus might indicate an additional therapeutic target. Moreover, in the age group 30 to ≤49 years (n = 11), we observed a receptor discordance for HER2 in 82% of the cases and for ER in 55% of the cases. In comparison, in the age group over 50 years (n = 18), we observed HER2 and ER discordance in 78% and 33% of cases, respectively. In contrast, HER2 discordant rates appeared to increase slightly with progressing menopausal status. While 75% of premenopausal patients presented with HER2 discordance, during and after menopause the rate increased to 80% and 83% of patients, respectively. In contrast, ER discordance decreased with age and menopausal status. Premenopausal women presented an ER discordance in 50% of the cases, while perimenopausal and postmenopausal patients showed ER discordance in 40% and 33% respectively. Tumor stage did not play a major role concerning ER and HER2 discordant rates. At the time of diagnosis, 18 patients presented with a T1 tumor (n = 18). Of these, 78% were HER2- and 44% were ER-discordant when looking at the DTCs. Similarly, patients with tumors bigger than 2 cm (T2 to T4, n = 11) showed receptor discordance for HER2 and ER in 82% and 36%. 

### 3.8. Investigation of DTC Clusters

We detected DTC cell clusters in six patients (21%), with a median of 1.4 clusters per patient. Clusters were defined as groups of two or more DTCs. Five of the six patients presented with a predominantly epithelial DTC phenotype. Of the six patients with cell clusters, three had luminal A, while the other three were diagnosed with luminal B/HER2+ primary tumors. Furthermore, five of the six patients (84%) revealed a discordance between the HER2 status of the primary tumor vs. DTCs in the bone marrow. Among the detected DTC clusters, we found the following profiles: 1× CK+, 2× CK+Ki67+, 2× HER2+, 1× CK+ER+, 1× CK+CD133+HER2+Ki67+, 1× CK+CD133+HER2+Ki67+Ncad+ (Figure 9). In four of the patients, we found one cluster each: CK+Ki67+ (ID 130; luminal B/HER2+); CK+ (ID196; luminal B/HER2+); CK+ER+ (ID 757; luminal A) and CK+Ki67+ (ID 1559; luminal A). One of the patients (ID 535; luminal B/HER2+) revealed two clusters, both with the profile CK+CD133+HER2+Ki67+Ncad+. In another patient (ID 1116; luminal A), we found two clusters, both with the profile HER2+. This patient was diagnosed with bone metastasis four months after primary diagnosis. Spearman’s Rho correlation showed that patients with a HER2-positive primary tumor were more likely to present DTC cell clusters in their bone marrow compared with patients with a HER2-negative primary tumor (r = 0.37, *p* = 0.048). The detection of clusters occurred more frequently in patients who had received neoadjuvant chemotherapy or targeted therapy (two of three patients) compared with patients receiving endocrine therapy (two of eight patients, r = 0.38, *p* = 0.045). 

### 3.9. DTC Subtype Detection vs. Standard Detection of CK+ Cells and Clinical Follow-Up

We detected 12 CK+CD133+HER2+Ki67+Ncad+ DTCs among nine patients whereof seven had luminal A tumors. Noteworthily, one of the patients (ID 1583) developed a glioblastoma during follow-up. At primary surgery, we detected four DTCs in 4 × 10^6^ bone marrow cells using the standard brightfield method solely based on CK ^23^. In contrast, we detected 10 DTCs among 2 × 10^6^ bone marrow cells using the sequential immunofluorescent staining procedure including 4× CK+, 2× CK+CD133+, 2× CK+CD133+Ki67+, 1× CK+CD133+Ki67+Ncad+, 1× CK+CD133+Her2+Ki67+Ncad+. The patient received endocrine therapy (Letrozole) but did not use adjuvant treatment with bisphosphonates. After 6 months, the patient was diagnosed with a glioblastoma. Another of these seveb patients (ID1116) had 28 CK-positive DTCs using the standard brightfield method. Similarly, using the immunofluorescent profiling, we detected 21 cells with CK alone, 3× CK+CD133+Ki67+, 3× CK+CD133+Ki67+Ncad+ and 1× CK+CD133+HER2+Ki67+Ncad+. This patient also presented with two DTC clusters. Remarkably, this patient did not use adjuvant bisphosphonate treatment and presented with primary bone metastasis four months after initial diagnosis. 

Notably, among the seven patients there was one with 24 CK-positive DTCs as detected by the standard method. The immunofluorescent staining revealed only one DTC with the following profile: CK+CD133+HER2+Ki67+Ncad+. The patient (ID 1457) received endocrine therapy but no adjuvant treatment with bisphosphonates. She presented a carcinoid in the lung, which was detected using computed tomography (CT) of the chest and confirmed by image-guided puncture one month after initial diagnosis.

Among the entire cohort, we discovered seven CK+HER2+ DTCs in six luminal A patients. One of them (ID 1442) became recurrent during follow-up. Applying the standard method, we detected five CK-positive DTCs. Using the immunofluorescent profiling, we found only one DTC, but this revealed the profile CK+HER2+ although the patient was diagnosed with a luminal A tumor. Noteworthily, she did not receive adjuvant treatment with bisphosphonates. 

Furthermore, throughout the study, we detected only one CK+Ncad+ cell in only one patient (ID 876; luminal B/HER2+) next to one CK+ DTC. In comparison, using the standard method, we detected 14 CK-positive DTCs. The patient received neoadjuvant chemotherapy combined with targeted therapy directed against HER2 and adjuvant treatment with bisphosphonates. However, she developed brain metastasis within 21 months after primary surgery. 

Overall, we found only one CK+CD133+Ki67+ER+ next to one CK+ DTC in another patient with a luminal B/HER2+ tumor (ID 552). The standard method discovered 12 CK-positive DTCs. This patient became recurrent after 721 days and presented with bone metastasis. 

## 4. Discussion

In this study, we investigated DTC subpopulations by establishing a sequential staining method with releasable and bleachable antibody–fluorochrome-conjugates. Technical limitations occurred during employment of releasing enzymes and photobleaching steps as they resulted in improvable fluorochrome quenching rates of 94.2% and 84.6%, respectively. However, with the aid of adequate control cells, sequential detection of the six cell-type-specific markers Pan-CK, CD133, Ki67, HER2, N-cadherin and ER was possible. In the literature, double immunofluorescence staining of DTCs was frequently reported using markers such as CK, HER2, ER, and CD45 [26,27,28,29,30,31]. To the best of our knowledge, this is the first time a sequential approach has been applied for DTC characterization [32,33]. Using sequential multi-parameter imaging, we were able to identify distinct DTC profiles not solely based on epithelial features. The detected DTCs showed tremendous heterogeneity as well as receptor discordance compared with the primary tumor. 

Among the 29 bone marrow samples from patients with HR+ breast cancer, we quantified a total number of 153 DTCs. The majority of the detected DTCs were CK-positive (128/153) and, hence, the most common DTC phenotype was the epithelial phenotype (in 55% of the patients). Since most publications concerning DTCs in breast cancer report the dichotomized DTC status (DTCs present or absent), it was difficult to compare the detected cell counts with other studies. However, we found a few studies reporting quantification of CK-positive DTCs and could confirm that our data was in a similar range [34,35].

Our DTC profiling approach indicated a connection between the occurrence of Ki67 and HER2 as well as to CD133. Particularly in CK-positive DTCs, the expression of CD133 appeared to occur together with Ki67. Further, N-cadherin expression was linked to CD133 and Ki67 positivity while HER2-positive cells were associated with the CD133+/Ki67+/Ncad+ profile. This association was partially reflected in the patients follow-up data. The events were elaborately described in the results section. The six patients that suffered from progression or recurrence presented CK-positive DTCs. Additionally, we detected CD133, HER2, Ki67, and N-cadherin in various combinations with CK. The quantity of DTCs detected with the standard brightfield method vs. the sequential immunofluorescence staining procedure differed: 4 vs. 10; 28 vs. 28; 24 vs. 1, 5 vs. 1; 14 vs. 2 and 12 vs. 2. Even though the detected number of DTCs was low using the sequential approach, the detected subpopulations appeared to be of prognostic relevance. 

Interestingly, we detected CK-negative DTCs (25/153) in 41% of the cohort. Here, our data indicated a connection between the epithelial marker CK and proliferation marker Ki67, as 76% of CK-negative DTCs were Ki67-positive. In contrast, only 53% of CK-positive DTCs showed Ki67 positivity, suggesting a potential for less aggressive behavior in CK-positive compared with CK-negative DTCs, as the expression of Ki67 antigen is known to be closely related to biologically aggressive tumor growth [36,37]. For the time being, there are no publications concerning CK-negative DTCs in breast cancer patients. The concept of EMT, which causes the loss of epithelial-like polarity and cell–cell adhesion potential, has been addressed in multiple studies focusing on circulating tumor cells (CTCs) in breast cancer patients, however [38,39]. On the one hand, EMT was reported to promote unfavorable outcomes in breast cancer patients [40,41]. Particularly, HER2+ DTCs appeared to play a key role, as they were associated with enhanced metastasis, EMT, stemness features, and invasive potential ^2^. On the other hand, transition of early tumor cells and cancer stem cells was reported without complete loss of the epithelial features [2,20]. In our study, among the CK-negative DTCs, the majority were HER2-positive or positive for CD133/HER2/Ki67/Ncad. 

Our data tally with the findings in previous studies reporting that DTCs from patients with HER2+ early breast cancer displayed unique gene expression patterns compared with invasive lesions and metastatic breast cancer patients. Interestingly, compared with CTCs in metastatic patients, the DTCs in the bone marrow of early HER2-positive breast cancer patients presented increased expression of stem cell markers such as ALDH1, CAV1 and VIM [42]. In contrast, we used CD133 instead of ALDH1 as a stemness marker, which was reported to interact with the Wnt/β-catenin and PI3K-Akt signaling pathways and to be associated with therapy resistance [43]. As an alternative to vimentin, which is also expressed by hematopoietic cells and thus might be troublesome to detect on DTCs in bone marrow, we chose N-cadherin, which was shown to be upregulated during EMT [40]. N-cadherin was reported to be of particular interest in the osteogenic niche, as bone metastases exhibited mesenchymal characteristics [41,44]. 

Using our DTC profiling approach, we found a connection between the proliferation marker Ki67 and HER2 in the HR+ cohort. Here, features of dormancy might play a role. HER2 positivity was correlated with Ki67-positive DTCs, indicating that they were less dormant and might re-awake to seed metastasis. In turn, ER-positive DTCs were negatively correlated with Ki67 DTCs, suggesting a more dormant state. These findings are in line with a study by Kim et al, who observed that an increase in dormancy scores as determined by gene signatures in various cell lines was correlated with a change from basal type to luminal breast cancer subtypes [45]. In our study, we observed that DTCs from patients with ER-positive primary tumors presented HER2. Further, the HER2-positive DTCs showed increased Ki67 positivity, which equals the luminal B subtype and could be interpreted with less dormancy compared with luminal A. Additionally, we observed that ER-positive DTCs were associated with the stem-cell-like DTC phenotype, thus suggesting a more dormant status. A stem-cell-like DTC phenotype in patients with early breast cancer was already reported by Balic et al [30]. In contrast to CD133, they used immunocytochemical staining against CD44 and CD24 in addition to CK. Previous studies reported that patients with HR+-positive primary tumors can be treated successfully, but often suffer from late recurrence [36]. In contrast, our data displayed that an epithelial DTC phenotype was associated with proliferation in terms of elevated Ki67-positive DTC counts suggesting a less dormant phase. At present, the dormancy status of DTCs has barely been investigated. In a study by Borgen et al., NR2F1 was suggested as a potential dormancy marker to identify patients at risk of developing bone metastasis [46]. A study by Spiliotaki explored proliferation in CTCs in serial blood samples from patients with early (≤5 years disease-free) vs. metastatic (>5 years after surgery) breast cancer using Ki67 staining together with Pan-CK and apoptosis marker M30 [47]. They reported that the proliferation index was increased on relapse. Moreover, in a study by Harper et al., HER2 was suggested to activate early tumor cell dissemination and promote the formation of metastasis after a dormancy phase [48]. 

Remarkably, we detected a HER2 receptor discordance in 79% of the patients in our HR+ cohort. Surprisingly, patients with luminal A tumors had a higher likelihood to present HER2+ DTCs compared with luminal B/HER2− and luminal B/HER2+ subtypes, hence suggesting the occurrence of discordance between DTCs and the primary tumor profile. HER2 positivity of DTCs and CTCs with HER2-negative primary tumors has been described in the literature, as well as loss of HER2 status in DTCs of HER2-positive primary tumors [49,50,51]. To explain this phenomenon, two theories have been posed: either, the amplification of HER2 is already acquired during tumor progression and dissemination of tumor cells [52], or HER2-positive cells of the primary tumor have a greater tendency to split from the tumor and form metastases [49,51]. Noteworthily, 2 of the 12 patients with local lymph node involvement displayed HER2 discordance in terms of receptor loss in the nodal sample compared with the primary tumor biopsy. Recently, a HER2 discordance rate of 12% between core needle biopsies of axillary lymph node metastases and the primary tumors of mostly therapy-naïve breast cancer patients has been described by our group [53], strengthening the hypothesis that dissemination and transformation of receptor status are early events in tumor progression. In line with this, tumor size was not a significant factor for HER2 or ER discordance rates concerning DTCs among our HR+ cohort. Furthermore, we detected ER discordance between DTC and primary tumor tissue in 41% of all cases and 75% in patients receiving neoadjuvant endocrine therapy. This coincides with a study by Fehm et al. reporting an ER concordance rate of only 28% (12 of 88 patients) between CK-positive DTCs and the primary tumor in patients diagnosed with primary breast cancer [31]. Although not significantly, ER discordance decreased with age and menopausal status in our cohort. In turn, HER2 discordance was slightly increased in post vs. pre-menopausal patients. A possible reason might be an impaired bone micro-environment as found in preclinical osteoporosis, which was shown to be correlated with postmenopausal status and could constitute a DTCs homing site [54]. 

Our study revealed the occurrence of DTC clusters in 6 of 29 patients. Interestingly, the clusters consisted of mainly epithelial DTCs. We did not find any publications addressing DTC clusters in breast cancer patients. Hence, we cannot compare our findings to other studies. It was, however, reported that CTC clusters were associated with a higher metastatic potency [39]. 

At the time being, there is no approved clinical test for DTC detection. In contrast, the Cell Search^®^ system was approved and recommended for CTC detection in patients with metastatic breast cancer, but no therapy decisions based on CTC phenotypes. Particularly in early breast cancer, CTC concentrations were reported to be very low [55]. The clinical use of CTC quantification was shown in metastatic breast cancer [56]. Although bone marrow aspiration for DTC detection appears to be more invasive compared with taking a peripheral blood sample for detection of CTCs, it might provide additional information. DTCs are an early event during the metastatic cascade, and their value as an independent prognostic factor for breast cancer recurrence has been well documented in past studies [4,57,58,59]. Furthermore, the adjuvant intake of bisphosphonates was shown to have an apoptotic effect on DTCs and to decrease the risk of bone metastases, hence providing an overall survival benefit [60,61,62] Thus, the investigation of DTCs in patients with primary breast cancer might become a clinically relevant tool.

Since our study is based on a very small case number, further investigations in a larger patient cohort are required to substantiate our data. 

## 5. Conclusions

In this study, we established and applied a sequential multi-parameter immunofluorescence staining procedure to investigate the phenotypical and therapy-related features of disseminated tumor cells in the bone marrow. We were able to detect distinct receptor profiles that differed enormously from the primary tumor tissue. These receptor discordances might have clinical implications and might be useful to monitor therapy response. Especially, patients with luminal A tumors revealed DTCs with potential therapeutic targets. We analyzed particular non-epithelial DTC subgroups and clusters that were never described before. Our findings indicate that characterization rather than quantification of DTCs might be relevant for prognosis and treatment decisions, making it possible to tailor more individual treatment approaches for patients in the future. 

Nevertheless, taking into account the non-randomized design and the small sample size, the results should be interpreted with proper caution. However, the findings of this study can be used as a hypothesis-generating basis, showing the need of further investigation of DTC characterization in studies with higher sample sizes. 

## Figures and Tables

**Figure 1 cancers-15-00568-f001:**
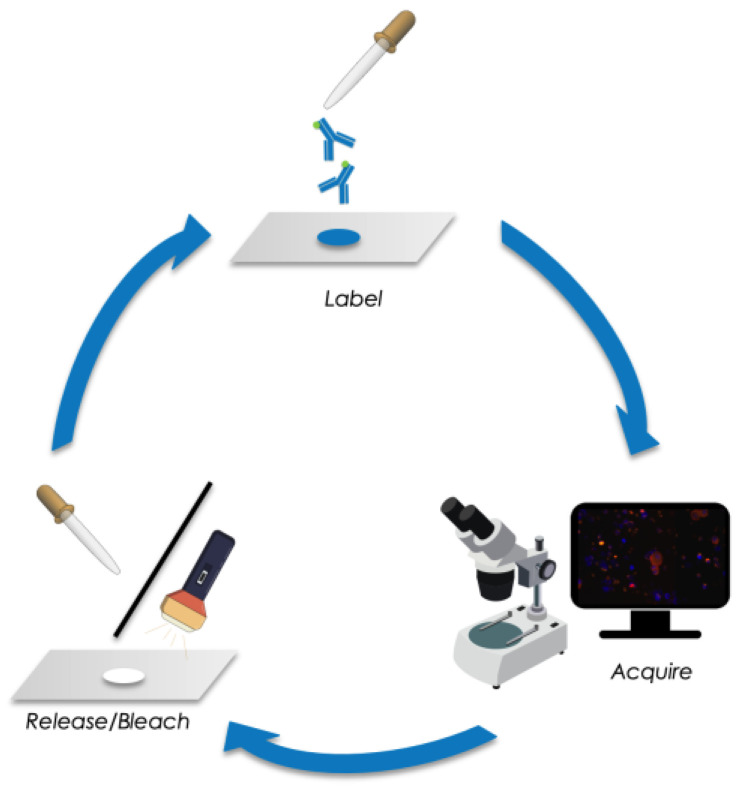
Schematic of sequential immunofluorescence staining followed by image acquisition and releasing/bleaching of fluorochromes.

**Figure 2 cancers-15-00568-f002:**
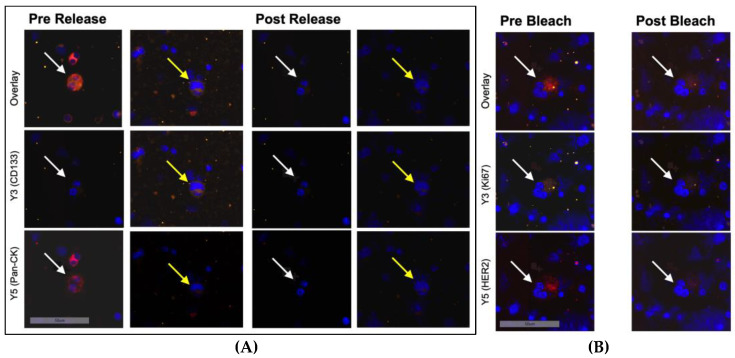
Immunofluorescent staining of ZR75-1 cells (white arrows) and T98G cells (yellow arrows) mixed with bone marrow cells at 20-fold magnification. (**A**) Before and after treatment with the releasing enzyme. Hence, secondary fluorescence of Pan-CK was successfully quenched and another set of antibodies could be applied. (**B**) Before and after bleaching with increased light exposure in channels Y3 and Y5. Shown is a loss of the secondary fluorescent activity of markers Ki67 and HER2 to prepare the sample for another set of antibodies.

**Figure 3 cancers-15-00568-f003:**
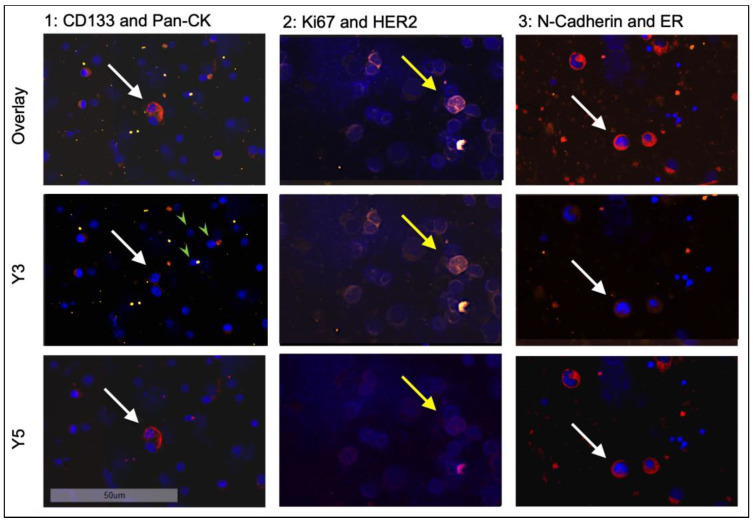
Sequential staining of positive control cell lines ZR75-1 (white arrows) and T98G (yellow arrows) at 20-fold magnification. Shown are three different tumor cells for each staining round. For comparison, hematopoietic bone marrow cells, which are distinctly smaller and only stain positive for DAPI, are marked with green arrowheads.

**Figure 4 cancers-15-00568-f004:**
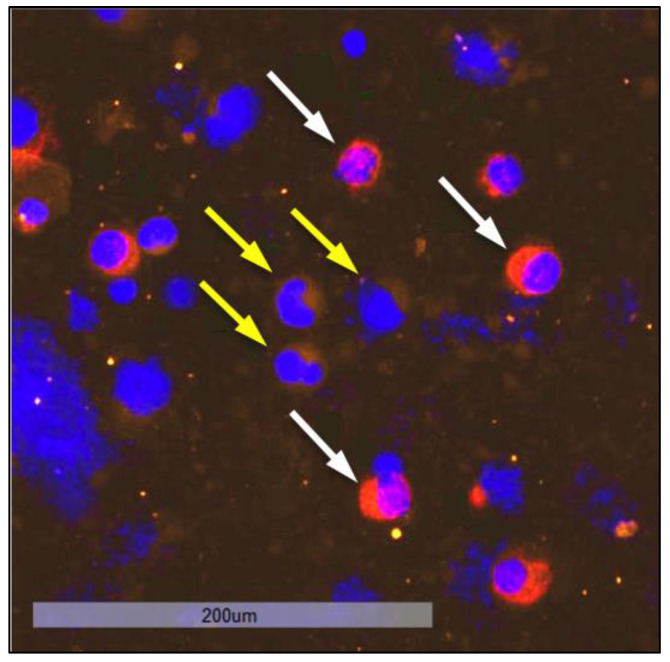
Staining of ZR75-1 cells (white arrows), which stain positive for ER (red) while remaining negative for N-cadherin, next to T98G cells (yellow arrows), which stain positive for N-cadherin (yellow) and negative for ER at 20-fold magnification.

**Figure 5 cancers-15-00568-f005:**
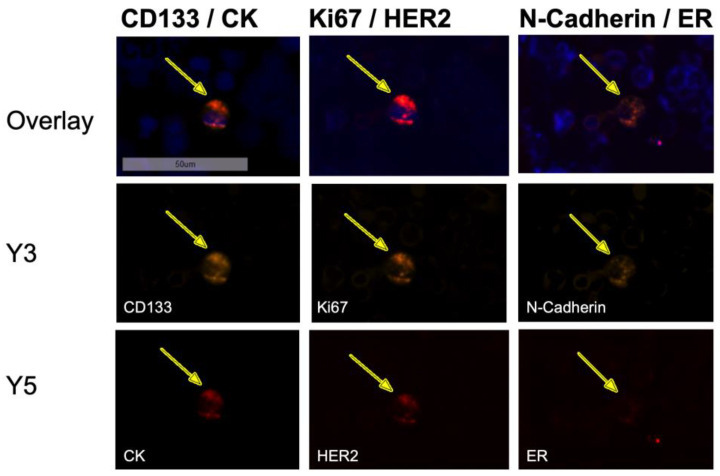
Sequential immunofluorescent staining of a disseminated tumor cell in the bone marrow of a patient exhibiting the profile CK+CD133+HER2+Ki67+Ncad+ at 20-fold magnification. The DTC expressed all stained markers (DAPI, CD133, CK, Ki67, HER2, N-cadherin) except ER, although the sample derived from a patient with an ER-positive primary tumor.

**Figure 6 cancers-15-00568-f006:**
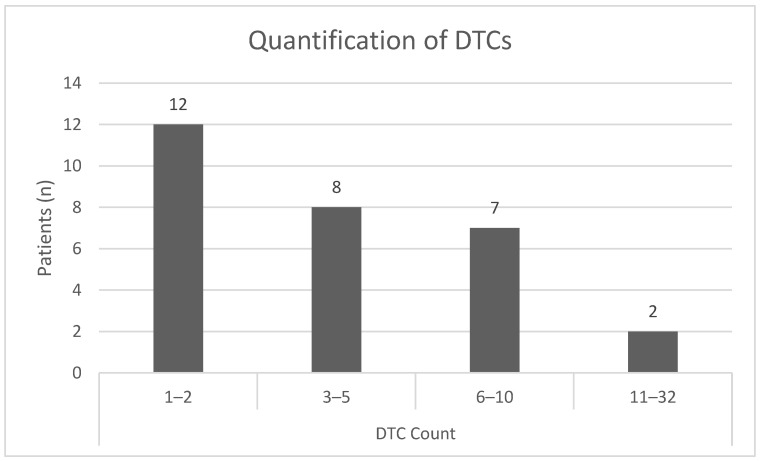
Quantification of DTCs per patient and division into subgroups defined by the number of DTCs detected in a bone marrow sample using sequential immunofluorescent staining.

**Figure 7 cancers-15-00568-f007:**
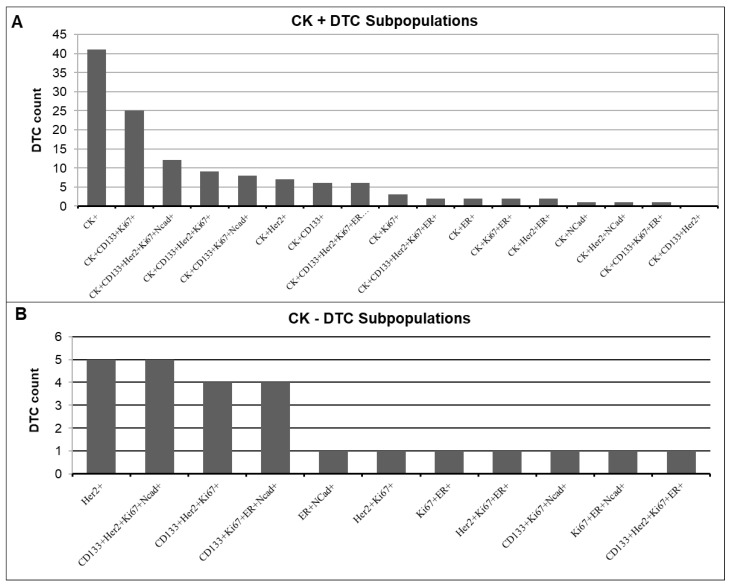
Quantification of DTC subpopulations in the (**A**) CK-positive and (**B**) CK-negative DTC subpopulations.

**Figure 8 cancers-15-00568-f008:**
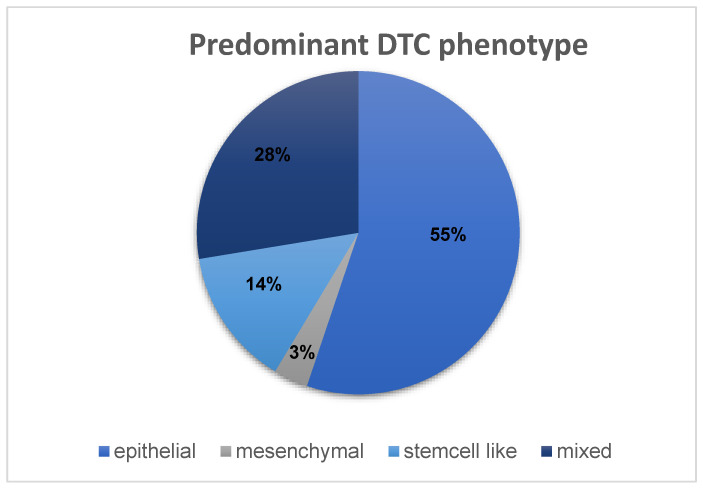
Distribution of DTC phenotypes among the HR+ cohort of 29 patients.

**Figure 9 cancers-15-00568-f009:**
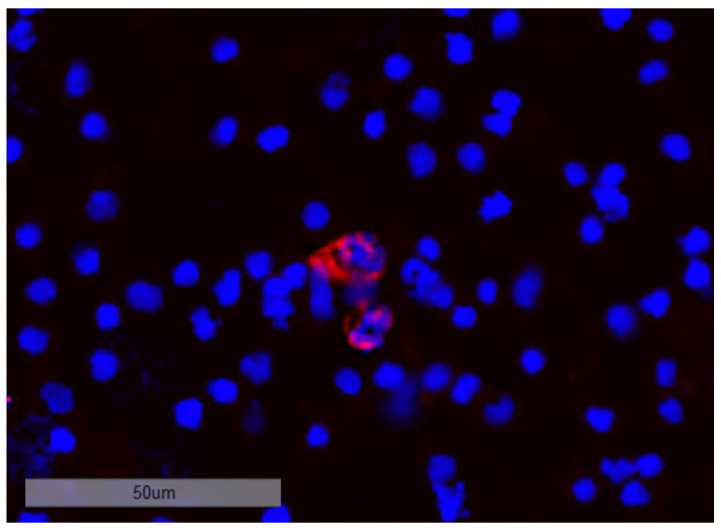
DTC cluster consisting of two cells staining positive for CK at 20-fold magnification.

**Table 1 cancers-15-00568-t001:** Patient characteristics and clinical parameters of primary tumor tissue.

Clinical Parameter	n (n = 29)	%
**Age**		
30 to ≤49	11	38%
50 to ≤64	11	38%
65 to ≤74	4	14%
≥75	3	10%
**Gender**		
female	28	97%
male	1	3%
**Molecular subtypes**		
Luminal A	17	59%
Luminal B/ HER2−	6	21%
Luminal B/ HER2+	6	21%
**Tumor size**		
pT1mic to T1c	18	62%
pT2	7	24%
pT3 to pT3b	3	10%
pT4d	1	3%
**Lymph node status**		
N0	17	59%
N1	7	24%
N2	3	10%
N3	2	7%
**Neoadjuvant therapy**		
No neoadjuvant therapy	18	62%
Chemotherapy combined with targeted antibodies	3	10%
Endocrine therapy	8	28%
**Ki67 in %**		
≤15	19	66%
16–49	7	24%
≥50	3	10%
**Menopausal status**		
pre-	12	41%
peri-	5	17%
post-	12	41%

**Table 2 cancers-15-00568-t002:** Immunofluorescent staining of cell lines ZR75-1 and T98G as well as hematopoietic bone marrow cells for the six different markers used.

Cells	ZR75-1	T98G	Hematopoietic Cells
Pan-CK	positive	negative	negative
CD133	negative	positive	negative
HER2	positive	negative	negative
Ki67	positive	positive	negative
ER	positive	negative	negative
N-Cadherin	negative	positive	negative
DAPI	positive	positive	positive

**Table 3 cancers-15-00568-t003:** Quantification of DTCs according to intrinsic subtype.

Pat.-ID	Subtype * Tumor	Total DTCs	Total CK+ DTC	Total HER2+ DTC	Total Ki67+ DTC	Total ER+ DTC	Total CD133+ DTC	Total Ncad+ DTC
243	1	6	6	0	3	0	3	0
275	1	5	4	2	3	0	5	0
277	1	11	7	8	10	1	10	3
536	1	2	2	2	2	1	2	2
757	1	5	3	2	2	1	0	0
775	1	7	4	1	3	5	3	4
956	1	6	3	5	4	2	3	2
1116	1	32	28	4	8	0	8	5
1124	1	8	7	6	5	1	5	2
1348	1	3	3	2	3	1	3	1
1363	1	3	3	2	0	0	0	0
1442	1	1	1	1	0	0	0	0
1457	1	2	1	1	2	1	1	2
1559	1	5	4	2	3	1	3	1
1583	1	10	10	1	4	0	6	2
1824	1	10	8	5	10	1	10	5
1864	1	1	1	1	0	0	0	0
**sum**	**17**	**117**	**95**	**45**	**62**	**15**	**62**	**29**
**mean**		**6.88**	**5.59**	**2.651**	**3.65**	**0.88**	**3.64**	**1.71**
542	2	5	3	0	4	2	3	3
563	2	1	1	1	1	1	1	0
878	2	2	2	0	1	0	1	0
1049	2	2	2	1	1	0	2	0
1519	2	1	1	0	1	1	0	0
1930	2	9	8	4	9	1	8	3
**sum**	**6**	**20**	**17**	**6**	**17**	**5**	**15**	**6**
**mean**		**3.33**	**2.83**	**1.00**	**2.83**	**0.83**	**2.50**	**1.00**
130	3	5	5	0	1	0	0	0
196	3	1	1	0	0	0	0	0
535	3	4	4	3	4	1	3	2
552	3	2	2	0	1	1	1	0
763	3	2	2	2	2	2	2	2
876	3	2	2	0	0	0	0	1
**sum**	**6**	**16**	**16**	**5**	**8**	**4**	**6**	**5**
**mean**		**2.67**	**2.67**	**0.83**	**1.33**	**0.67**	**1.00**	**0.84**
Total	29	153	128	56	87	24	83	40

* 1 = luminal A; 2 = luminal B/HER2−; 3 = luminal B/HER2+.

**Table 4 cancers-15-00568-t004:** Quantification of CK-positive and CK-negative DTC subpopulations.

	**Ki67+ CK+**	**HER2+ CK+**	**CD133+ CK+**	**ER+ CK+**	**Ncad+ CK+**
n DTCs	68	39	69	15	28
% of CK+ DTCs	53	30	54	12	22
% of total DTCs	44	25	45	10	18
	**Ki67 + CK−**	**HER2 + CK−**	**CD133+ CK−**	**ER+ CK−**	**Ncad+ CK−**
n DTCs	19	17	15	9	12
% of CK- DTCs	76	68	60	36	48
% of all DTCs	12	11	10	6	8

**Table 5 cancers-15-00568-t005:** Receptor discordance between DTCs and tumor tissue in patients without therapy, compared with patients receiving neoadjuvant endocrine therapy or neoadjuvant chemotherapy combined with targeted therapy.

Receptor	No Neoadjuvant Therapy n (%)	Endocrine Therapy n (%)	Chemo-/Targeted Therapy n (%)
Discordance ER	8 (44%)	2 (25%)	2 (66%)
Concordance ER	10 (56%)	6 (75%)	1 (33%)
Discordance HER2	14 (78%)	7 (87,5%)	2 (66%)
Concordance HER2	4 (22%)	1 (12,5%)	1 (33%)

## Data Availability

The data presented in this study are available on request from the corresponding author.

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
