# Peer review of "Multi-Parameter Analysis of Disseminated Tumor Cells (DTCs) in Early Breast Cancer Patients with Hormone-Receptor-Positive Tumors"

_cancers, 2023, doi:10.3390/cancers15030568_

Round 1

Reviewer 1 Report

In this manuscript, Konig and co-workers report investigation of disseminated tumor cells (DTC) in a cohort of 29 estrogen receptor (ER) positive breast cancer patients at the time of surgery. At the time of sampling bone marrow, a minority of the patients were therapy naïve while the majority had received i) neoadjuvant chemotherapy (n=11), or ii) hormonal treatment (n=8), or HER2-targeted therapy (n=3)

The investigation was thoroughly conducted, and the manuscript reports novel technical aspects that are of interest. The authors acknowledge limitations of their work (for technical aspects, lines 466-470, for generalizability of data lines 625-628)

Major observations

The discussion is lengthy and from line 490 to line 526 reports results, with particular reference to discrepancies between CK+ DTC detection with the brightfield standard method and the sequential multiparameter staining. This reviewer invites the authors to consider transferring these data in the Results section, and discuss them in the Discussion section

Also, since quantification and characterization of DTC requires bone marrow aspirates, the authors should discuss if and to what extent their analysis of DTC provides additional or complementary clinically valuable information as compared to readily available circulating tumor cells

Minor observations

Page 2 line 71 “tripe” should be “triple”

Page 4 line 121 with reference to the “prior positive detection of DTCs”, this reviewer takes it that from the same bone marrow aspirate DTC were quantified with both a standard brightfield method using an anti-CK monoclonal antibody and the  sequential multiparameter staining

Page 7 there is no reference in the body of the article to figures 3 and 4

Page 9 lines 293-294 the article reads 9 patients but there are 7 luminal A, 2 luminal B/HER2-ve and 2 luminal B/HER2+ve

Page 12 Table 3 this reviewer’s suggestion is to swap CK-ve and CK+ve data so that they will be displayed in the same order as in figure 7

Page 19 lines 649-654 “Informed Consent Statement”, “Acknowledgments” and “Conflict of Interest” should be removed

Author Response

Point-by-Answers to Reviewers’ questions:

Reviewer 1:

The discussion is lengthy and from line 490 to line 526 reports results, with particular reference to discrepancies between CK+ DTC detection with the brightfield standard method and the sequential multiparameter staining. This reviewer invites the authors to consider transferring these data in the Results section, and discuss them in the Discussion section.

We thank the reviewer for the constructive criticism. As suggested, we moved the paragraph describing DTC subtypes compared to the standard method under the aspect of clinical follow up from the discussion to the results section. New subheading : DTC subtype detection vs. standard detection of CK+ cells and clinical follow up. Further, we added a short paragraph in the discussion concerning the comparison of both detection methods: The 6 patients that suffered from progression or recurrence presented CK positive DTCs.  Additionally we detected CD133, HER2, Ki67 and N-cadherin in various combinations with CK. The quantity of DTCs detected with the standard brightfield method vs. the sequential immunofluorescence staining procedure differed: 4 vs. 10; 28 vs. 28; 24 vs. 1, 5 vs. 1; 14 vs. 2 and 12 vs. 2. Even though the detected number of DTCs was low using the sequential approach, the detected subpopulations appeared to be of prognostic relevance.

Also, since quantification and characterization of DTC requires bone marrow aspirates, the authors should discuss if and to what extent their analysis of DTC provides additional or complementary clinically valuable information as compared to readily available circulating tumor cells.

This is an important remark. We added a paragraph at the end of the discussion section concerning our reasons for analyzing DTCs from bone marrow rather than CTCs form peripheral blood.:

At the time being, there is no clinical test for DTC detection. In contrast, the Cell Search® system was approved and recommended for CTC detection in patients with metastatic breast cancer, but no therapy decisions based on CTC phenotypes. Particularly in early breast cancer, CTC concentrations were described to be very low [55]. The clinical use of CTC quantification was shown in metastatic breast cancer [56]. Although bone marrow aspiration for DTC detection appears to be more invasive compared to taking a peripheral blood sample for detection of CTCs, it might provide additional information. DTCs are an early event during the metastatic cascade, and their value as an independent prognostic factor for breast cancer recurrence has been well documented in past studies [57–59, 4]. Furthermore, the adjuvant intake of bisphosphonates was shown to have an apoptotic effect on DTCs and to decrease the risk of bone metastases thus, providing an overall survival benefit [60–62]. Thus, the investigation of DTCs in patients with primary breast cancer might become a clinically relevant tool.

Minor observations

Page 2 line 71 “tripe” should be “triple”: corrected.

Page 4 line 121 with reference to the “prior positive detection of DTCs”, this reviewer takes it that from the same bone marrow aspirate DTC were quantified with both a standard brightfield method using an anti-CK monoclonal antibody and the  sequential multiparameter staining: clarified.

Page 7 there is no reference in the body of the article to figures 3 and 4: amended

Page 9 lines 293-294 the article reads 9 patients but there are 7 luminal A, 2 luminal B/HER2-ve and 2 luminal B/HER2+ve: corrected

Page 12 Table 3 this reviewer’s suggestion is to swap CK-ve and CK+ve data so that they will be displayed in the same order as in figure 7: applied

Page 19 lines 649-654 “Informed Consent Statement”, “Acknowledgments” and “Conflict of Interest” should be removed: applied

Reviewer 2 Report

In the manuscript by König et al., the authors applied a novel, sequential approach for DTC identification and characterization. They present a method based on sequential multi-parameter imaging, which enabled them to identify distinct DTC profiles. In turn, analysis of the detected DTCs revealed a notable heterogeneity, as well as receptor discordance compared to the primary tumor. Overall, the described method, as well as the presented results are novel and interesting. There are, though, a few points that need to be further processed before acceptance:

1.     Due to the plethora of antibodies and cells used in the present study, a table including all examined cells/ cell lines and positive and negative stains would be really useful in the Results section.

2.     In the Results section, in the paragraph “Identification and quantification of DTC-Subtypes in HR+ patients” the DTC positive/ negative stains should be more clearly presented.

3.     Since there is a great number of other commonly used DTC markers, such as TWIST1, SCGB2A2, MUC 1, etc., the authors should describe a clear rationale, why they chose these specific markers for DTC identification.

4.    The authors have performed a sequential multi-parameter imaging method based on immunofluorescence staining for the identification and characterization of DTCs. As a suggestion, a representative portion of samples, at least for some essential markers, should be confirmed by another method e.g., PCR. Such data would further strengthen the presented results.

Author Response

Reviewer 2:

In the manuscript by König et al., the authors applied a novel, sequential approach for DTC identification and characterization. They present a method based on sequential multi-parameter imaging, which enabled them to identify distinct DTC profiles. In turn, analysis of the detected DTCs revealed a notable heterogeneity, as well as receptor discordance compared to the primary tumor. Overall, the described method, as well as the presented results are novel and interesting. There are, though, a few points that need to be further processed before acceptance:

We appreciate the reviewer’s positive evaluation of our work and are thankful for the useful comments.

Due to the plethora of antibodies and cells used in the present study, a table including all examined cells/ cell lines and positive and negative stains would be really useful in the Results section.

This was a very helpful suggestion. We added a new Table 2 to the results section and hope that the positive and negative staining signals became clearer now. The numbering of tables has been adjusted.

In the Results section, in the paragraph “Identification and quantification of DTC-Subtypes in HR+ patients” the DTC positive/ negative stains should be more clearly presented.

                We rearranged figure 5 and hope that the positive and negative stains become clear now.

Since there is a great number of other commonly used DTC markers, such as TWIST1, SCGB2A2, MUC 1, etc., the authors should describe a clear rationale, why they chose these specific markers for DTC identification.

This is a valuable point. We added a sentence concerning the choice of our antibodies in the “Sequential immunofluorescence staining and DTC imaging” section. However, we question whether the suggested markers TWIST1, SCGB2A2 and MUC 1 are solely DTC markers. When conducting literature research, we found that double immunofluorescence staining of DTCs was frequently reported using markers such as CK, HER2, ER and CD45. To the best of our knowledge, there was no previous sequential staining approach for DTC characterization. Further, we did not find any publications concerning CK negative DTCs in breast cancer patients, yet. Moreover, we found that the DTCs in the bone marrow of early HER2 positive breast cancer patients presented increased expression of stem cell markers such as ALDH1, CAV1 and VIM. Because it was previously associated with therapy resistance, we preferred to use CD133 as a stemness marker. As an alternative to vimentin, which is also expressed by hematopoietic cells and thus might be troublesome to detect on DTCs in bone marrow, we chose N-cadherin, which was shown to be upregulated during EMT. N-cadherin was described to be of particular interest in the osteogenic niche, as bone metastases exhibited mesenchymal characteristics according to the literature. We explained this in the discussion section.

The authors have performed a sequential multi-parameter imaging method based on immunofluorescence staining for the identification and characterization of DTCs. As a suggestion, a representative portion of samples, at least for some essential markers, should be confirmed by another method e.g., PCR. Such data would further strengthen the presented results.

This is certainly a great idea. Actually, we attempted to performed single cell PCR on 10 bone marrow samples. After locating the cells on the slides, we picked them. One problem was the quality of the cells though. Another problem was the low DNA concentration, which made it necessary to pool cells. Hence, we could not gain sufficient results and changed our procedure.  In order to pick DTCs from a slide, the cytospin is not a suitable method. We perform an additional cytespread rather. The procedure is, however, very pricey and time consuming. Currently, we are collecting cytespreads (selected patients only) and are planning on analysis of batches.

Round 2

Reviewer 1 Report

Both quality of presentation and discussion of results improved in this revised version

This reviewer has no additional comments and congratulates the authors on this interesting manuscript

Reviewer 2 Report

NONE